# Immune Checkpoint Inhibitors in Urothelial Carcinoma: Recommendations for Practical Approaches to PD-L1 and Other Potential Predictive Biomarker Testing

**DOI:** 10.3390/cancers13061424

**Published:** 2021-03-20

**Authors:** Antonio Lopez-Beltran, Fernando López-Rios, Rodolfo Montironi, Sophie Wildsmith, Markus Eckstein

**Affiliations:** 1Department of Pathology and Surgery, Faculty of Medicine, Cordoba University, 14004 Cordoba, Spain; 2Faculty of Medicine, Champalimaud Clinical Center, 1400-038 Lisbon, Portugal; 3Pathology-Targeted Therapies Laboratory, HM Hospitales, 28050 Madrid, Spain; flopezrios@hmhospitales.com; 4School of Medicine, Polytechnic University of the Marche Region (Ancona), 60126 Ancona, Italy; r.montironi@staff.univpm.it; 5AstraZeneca R&D Oncology, AstraZeneca, Cambridge SG8 6EH, UK; sophie.wildsmith@astrazeneca.com; 6Institute of Pathology, University Hospital Erlangen, Friedrich-Alexander-Universität Erlangen-Nürnberg, 91054 Erlangen, Germany; markus.eckstein@uk-erlangen.de

**Keywords:** assays, best practice, biomarkers, PD-L1, urothelial carcinoma

## Abstract

**Simple Summary:**

The predominant histologic type of bladder cancer is urothelial carcinoma (UC). Programmed cell death-ligand 1 (PD-L1) expression levels in UC tumors help clinicians determine which patients are more likely to respond to immuno-oncology (IO) therapies; as such, the harmonization of PD-L1 testing in evaluating patients is increasingly important. A series of international workshops, involving renowned pathologists and oncologists, were held to develop best practice approaches to PD-L1 testing in UC. It was agreed that robust control of analytical standards is required to obtain quality PD-L1 results and that interpretation and reporting of PD-L1 require clear inter-clinician communication. Recommendations for the best practices for PD-L1 testing in UC are provided. A PD-L1 test request form for pathology laboratories was also developed and included here, encouraging communication between clinicians and pathologists, and ensuring fast and high-quality test results. Novel biomarkers being evaluated for immuno-oncology agents in UC are also briefly discussed.

**Abstract:**

Immuno-oncology (IO) agents (anti–programmed cell death 1 (PD-1) and anti–programmed cell death-ligand 1 (PD-L1)) are approved as first- and second-line treatments for metastatic UC. PD-L1 expression levels in UC tumors help clinicians determine which patients are more likely to respond to IO therapies. Assays for approved IO agents use different antibodies, immunohistochemical protocols, cutoffs (defining “high” vs. “low” PD-L1 expression), and scoring algorithms. The robust control of pre-analytical and analytical standards is needed to obtain high-quality PD-L1 results. To better understand the status and perspectives of biomarker-guided patient selection for anti–PD-1 and anti–PD-L1 agents in UC, three workshops were held from December 2018 to December 2019 in Italy, Malaysia, and Spain. The primary goal was to develop recommendations for best practice approaches to PD-L1 testing in UC. Recommendations pertaining to the interpretation and reporting of the results of PD-L1 assays from experienced pathologists and oncologists from around the globe are included. A test request form for pathology laboratories was developed as a critical first step for oncologists/urologists to encourage communication between clinicians and pathologists, ensuring fast and high-quality test results. In this era of personalized medicine, we briefly discuss novel biomarkers being evaluated for IO agents in UC.

## 1. Introduction

The predominant histologic type of bladder cancer is urothelial carcinoma (UC), accounting for more than 90% of all cases [1]. UC is a heterogeneous disease that has historically been categorized into two distinct entities based on histologic features and genetic alterations—papillary non–muscle-invasive tumors (arising from epithelial hyperplasia sequence) and muscle-invasive tumors (arising from a dysplasia/carcinoma in situ sequence). However, more recently, comprehensive RNA expression profiling studies have identified at least five subtypes, the most fundamental division being basal/squamous-like and luminal [2]. Approximately 70% of patients are diagnosed with non–muscle-invasive tumors that infrequently metastasize, whereas 25% of patients are diagnosed with muscle-invasive tumors, of which approximately 40% will metastasize; 5% of patients are initially diagnosed with metastatic disease [3].

Although UC is mostly observed in its pure or conventional form, several histologic variants of invasive disease are now recognized by the World Health Organization, which arise from the propensity of UC for divergent differentiation [1,4]. While it is not completely understood how the different variants of UC affect prognosis, a greater understanding of the biology of UC has led to breakthrough treatments for metastatic disease. In particular, immuno-oncology (IO) agents (anti–programmed cell death 1 (PD-1) and anti–programmed cell death-ligand 1 (PD-L1)) have been studied as first- and second-line treatments for metastatic UC [5,6,7,8]. PD-L1 expression levels in UC tumors also help clinicians determine which patients are more likely to respond to IO therapies. Multiple assays approved for use with these IO agents use different antibodies, immunohistochemical (IHC) protocols, cutoffs (to define “high” vs. “low” PD-L1 expression), and scoring algorithms [9].

To understand better the current status and perspectives of biomarker-guided patient selection for anti–PD-1 and anti–PD-L1 agents in UC, three workshops were held from December 2018 to December 2019 in Milan, Italy; Kuala Lumpur, Malaysia; and Madrid, Spain. Here, we report the findings from these workshops, the primary goal of which was to develop a set of best practice recommendations for PD-L1 testing in UC. We include recommendations pertaining to the interpretation and reporting of the results of PD-L1 assays from renowned pathologists and oncologists from around the globe. Furthermore, in this era of personalized medicine, we briefly discussed novel biomarkers being evaluated for IO agents in UC and introduced the concept of developing a master protocol to integrate these biomarkers into UC treatment regimens in the future.

## 2. Current Treatment Landscape for UC

The immune sensitivity of bladder cancer has been recognized since the 1970s when intravesical bacillus Calmette–Guérin was discovered [10], which, along with transurethral resection of bladder tumor (TURB), remains today as the standard of care (SoC) for non–muscle-invasive disease in patients with high grade and/or T1 tumors [3], in patients with Ta high-grade tumors, mitomycin is an additional treatment option [11]. Treatment of muscle-invasive bladder cancer includes surgery (radical cystectomy) with or without neoadjuvant chemotherapy, and patients at high risk of recurrence may be treated with adjuvant chemotherapy [12]. For the first-line treatment of unresectable, metastatic UC, platinum-based chemotherapy has remained the SoC for more than two decades [13]. Eligible patients can receive cisplatin-based regimens, most commonly gemcitabine plus cisplatin. Patients unable to receive cisplatin (approximately 50%) often receive gemcitabine plus carboplatin [14].

It has been shown that the number of tumor-infiltrating lymphocytes is lower in more advanced, metastasizing UC [15,16,17,18], with lower numbers of CD4^+^ T cells being particularly associated with a poor prognosis [19]; thus, IO agents may be effective in restoring the immune response dampened by tumor immunosuppressive mechanisms. Indeed, the anti–PD-L1 agents atezolizumab, avelumab, and durvalumab and the anti–PD-1 agents pembrolizumab and nivolumab are currently, or were previously, approved by the US Food and Drug Administration (FDA) for second-line treatment of advanced UC [20,21,22,23,24]. The approvals are regardless of PD-L1 status [5], although US FDA- and Conformitè Europëenne (CE)-approved in vitro diagnostic device tests have been available since 2017 to inform physician decisions. The treatment landscape changed in 2017 when two of the IO agents were approved for the first-line treatment of metastatic UC in cisplatin-ineligible patients. Pembrolizumab [23] and atezolizumab were approved by the US FDA for the first-line treatment of cisplatin-ineligible patients with stage IV UC and high tumor PD-L1 expression, or for patients not eligible for any platinum-based therapy regardless of PD-L1 status; subsequent to accelerated approval, both therapies were restricted for use in PD-L1–high patients based on data from ongoing studies. The accelerated approvals were based on single-arm, phase II studies with objective response rate by response evaluation criteria in Solid Tumors version 1.1 as the primary endpoints [25,26]. The European Medicines Agency (EMA) has approved atezolizumab and pembrolizumab for first-line treatment of cisplatin-ineligible patients with stage IV UC and high tumor PD-L1 expression. More recently, avelumab combined with best supportive care (BSC) demonstrated an improvement in overall survival (OS) versus BSC alone in patients who had not progressed following chemotherapy for advanced UC. Patients who did not progress were enrolled in the trial. Survival was higher in patients with tumors expressing PD-L1, as measured using the VENTANA PD-L1 (SP263) Assay [27].

The efficacy of IO agents in metastatic UC has prompted investigators to evaluate the potential of these therapies in earlier stages of the disease, i.e., non–muscle-invasive tumors and in the neoadjuvant and adjuvant settings for localized muscle-invasive bladder cancer [28,29,30]. Collectively, there is increasing use of anti–PD-1 and anti–PD-L1 agents for the treatment of UC across disease stages, underscoring the role of PD-L1 testing in evaluating patients.

## 3. The Role of PD-L1 in UC

It is well established that PD-L1 on tumor cells (TCs) inhibits the antitumor responses of PD-1-expressing CD8^+^ T cells [31]. However, PD-L1 expression on immune cells (ICs) (e.g., tumor-associated macrophages) may also be important for the escape of TCs from immune detection [32], particularly for more immunogenic tumors such as UC [9]. The expression of PD-L1 on ICs appears to be of greater magnitude and of longer duration than on TCs and PD-L1 expression is IFNγ-dependent in TCs but only partially IFNγ-dependent in ICs [9,32], suggesting that the majority of PD-L1 in the immunosuppressive tumor microenvironment may be provided by ICs [32].

DNA (promoter) methylation regulates the expression of PD-L1 and is associated with interferon signaling transcriptional phenotype [33]. PD-L1 promoter methylation, resulting in down-regulation of expression, may be a mechanism of resistance to anti–PD-1 and anti–PD-L1 agents [34]. Several studies have evaluated the prognostic value of PD-L1 expression [9,16,35,36]. Mixed results have been reported, but most studies have shown a poorer prognosis for patients with high TC PD-L1 expression [16,37,38,39,40,41], and conversely, higher IC PD-L1 expression is associated with a better prognosis in metastatic UC [9,16,42]. The positive prognostic role of PD-L1 expression on ICs is supported by the outcome of phase III IMvigor130 trial; using an IC-only algorithm, patients with high PD-L1 (IC ≥ 5%) treated with both atezolizumab and SoC had higher survival than all-comers [43]. Conversely, in the KEYNOTE-052 trial, in which a combined TC and IC algorithm was used (combined positive score (CPS) ≥ 10), survival was lower in the patients with high PD-L1 than all-comers in both the pembrolizumab and SoC arms [25].

The role of PD-L1 in predicting outcomes with anti–PD-1 and anti–PD-L1 agents has yielded mixed results [9]. A meta-analysis of 10 studies with immune checkpoint inhibitors (prior to 2018) concluded that PD-L1 expression was associated with objective response rates but not OS [5]. More recently, the data from phase III randomized controlled trials have highlighted variability in the ability of PD-L1 to predict outcomes, particularly for IC-only approaches. For example, patients with high PD-L1 expression (IC ≥ 5%) in the IMvigor130 trial showed improved survival over those with low expression (median OS not estimable vs. 13.5 months) [44], whereas lower survival was seen in PD-L1–high patients than PD-L1 low/negative patients in IMvigor210 (median OS 12.3 months vs. 19.1 months) [26]. In the IMvigor10 trial, in high-risk patients with muscle-invasive bladder cancer (MIBC), PD-L1 expression did not enrich for disease-free survival (hazard ratio (HR) 1.01 vs. HR 0.81 in PD-L1 high vs. low/negatives) [28]. Using both TC and IC PD-L1 expression, the CPS assay predicted improved survival in pembrolizumab-treated patients in the KEYNOTE-045 (HR 0.73 vs. 0.57) [35] and KEYNOTE-052 trials (median OS 12.3 months vs. 18.5 months) [45]. Similarly, recent data from the JAVELIN Bladder 100 trial of maintenance therapy showed improved survival in the PD-L1 high versus all-comers (HR 0.56 vs. 0.69). This indicates that the PD-L1 assay with a TC/IC ≥ 25 algorithm is predictive, although expression data were incomplete [27]. The utility of PD-L1 for predicting response in non-small cell lung cancer has been demonstrated, mainly by assessing PD-L1 on TCs, the result of which indicated significant levels of expression. However, in UC, the expression on ICs predominates [46]. This differential pattern of PD-L1 expression may explain differences in predictive utility. There is also evidence that PD-L1 expression may be prognostic in UC, which has the potential to confound predictive effects [36].

## 4. Evaluation of PD-L1 in UC

Five validated, commercial assays are available to assess PD-L1 expression on TCs and ICs for approved anti–PD-1 and anti–PD-L1 agents (Table 1) [9,47,48], of which four are FDA approved (pharmDX 73-10 is not FDA approved; Table 1). These assays use different antibodies, IHC protocols, cutoffs (to define “high” vs. “low” PD-L1 expression), and scoring algorithms, the latter of which varies widely across assays in UC. Each algorithm and cutpoint was independently developed using clinical data from a therapy to best identify patients who would benefit; although the PD-L1 antibody stains may be similar, the algorithms are unique to each drug. Often, the cutpoints were determined using limited patient numbers, single-arm trials, and response data. Later in development, OS data may support the choice of the optimal cutpoint [49]. The different scoring algorithms for the companion diagnostic assays of validated or approved IO agents are summarized in Table 1 [50,51,52,53].

PD-L1 testing in conjunction with pembrolizumab in UC (PD-L1 IHC pharmDx 22C3 assay) is based on the combination of the total number of TCs and ICs with PD-L1 staining as a proportion of the total number of TCs (CPS); for TC scoring with nivolumab (PD-L1 IHC pharmDx 28-8 Assay), durvalumab (VENTANA PD-L1 (SP263) Assay), and avelumab (PD-L1 IHC pharmDx 73-10 Assay) the proportion of TCs with PD-L1 staining per TC area is assessed; with durvalumab (VENTANA PD-L1 (SP263) Assay) the proportion of ICs with PD-L1 staining within the total IC area is used; with atezolizumab (VENTANA PD-L1 (SP142) Assay), the proportion of the tumor area occupied by PD-L1 stained ICs is measured. It is important to note when reporting scoring that the method of scoring ICs is different for durvalumab (VENTANA PD-L1 (SP263) Assay) and atezolizumab (VENTANA PD-L1 (SP142) Assay). Examples of IC and TC PD-L1 staining in UC using the VENTANA PD-L1 (SP263) Assay are provided in Figure 1.

Preliminary findings suggest that it may be feasible to measure PD-L1 in circulating tumor cells (CTCs). However, these methods would not include PD-L1 expression on tumor-infiltrating ICs and the reproducibility and clinical utility of these assays remain to be proven [54].

## 5. Methodological Considerations

### 5.1. Pre-Analytics

Robust control of pre-analytical and analytical standards is needed to obtain high-quality PD-L1 results. Given the number of companion PD-L1 assays available, a test request form for pathology laboratories is a critical first step for oncologists/urologists [55]. The initial exchange of information between oncologists/urologists and pathologists is extremely important to determine the specific IO therapy for a given patient. We, therefore, recommend the use of the PD-L1 test request form that requires input from both clinicians and pathologists, as shown in Figure 2.

There are two ways to approach the scoring algorithm using the test request form, and the appropriate approach should be chosen for each institution. In the first approach, the clinician chooses an IO therapy and then informs the pathologist in order to match the scoring algorithm to the preferred IO therapy. Using the second approach, the pathologist would report the results of all scoring algorithms, and then the clinician makes an informed decision regarding the IO therapy based on the “best” scoring algorithm.

It should be noted that the differences in the approved assays mean that different PD-L1 classifications are likely, i.e., a patient with metastatic UC may be eligible for first-line treatment with one agent but not with another. For instance, discordance in the selection of populations between the VENTANA PD-L1 (SP142) Assay and the PD-L1 IHC pharmDx 22C3 Assay was reported in approximately 42% of patients who were PD-L1 positive by at least one of the assays [56], and fewer patients are deemed eligible for first-line treatment with an IO agent based on the SP142 assay [56]. An assay comparison study, in which UC tumor samples were stained and scored with the algorithm and cutpoint associated with that assay, demonstrated the overlap of positive cases using commercially available PD-L1 assays (Figure 3) [48].

Pathologists may have several sample types available for analysis, including biopsy (lymph nodes and metastatic tissue), TURB, and cystectomy specimens. In all cases, it is important to fix the specimen as soon as possible and within a maximum of 30 min after biopsy/resection. This may require significant coordination between the pathologist and surgeon, for example, for specimens obtained from resections or cystectomies. At present, there are limited data regarding differences in sample type for the assessment of PD-L1 using the available IHC assays. Based on our combined experience, our recommendations for the optimal conditions for PD-L1 immunohistochemistry are described in Table 2 [57,58].

### 5.2. Analytics

Challenges faced by pathologists for PD-L1 testing are both biological and technical. Biological considerations include intra-tumor heterogeneity within samples (particularly for small biopsies and TURB samples) exclusion of necrotic areas and tissue biomarker changes observed over time and in response to therapy.

Technical considerations include the appropriate use of the four FDA/EMA-approved PD-L1 assays that utilize different antibodies, platforms, and scoring algorithms [55].

Based on analytical studies for each assay, inter- and intra-observer agreement appear high. For the VENTANA PD-L1 (SP263) Assay (TC/IC ≥ 25%), inter- and intra-reader precision studies demonstrated overall, negative, and positive percent agreement (OPA, NPA, and PPA, respectively) of >90% [47]. For the VENTANA PD-L1 (SP142) Assay (IC ≥ 5%), in UC samples, the inter-reader precision OPA was 88.8% and the intra-reader precision OPA was 93.6% [59]. For the PD-L1 IHC pharmDx 22C3 Assay (CPS10), inter- and intra-reader OPA, NPA, and PPA were all >90% [60]. For the PD-L1 IHC pharmDx 28-8 Assay (TC ≥ 1%), inter and intra-reader OPA, NPA, and PPA were all >90% [61].

There are differences between assays in terms of staining patterns and the ability to detect different PD-L1-expressing cell types. Different assays present with particular staining patterns, e.g., the VENTANA PD-L1 (SP142) Assay with an ant-like/dot-like granular staining pattern or the PD-L1 IHC pharmDx 22C3 with an overall weak staining intensity (Table 1). However, staining characteristics or staining intensity are not included in PD-L1 scoring. More importantly, PD-L1 assays can show different performances in detecting proportions of different PD-L1-expressing cell types. All FDA-approved assays show comparable performance in detecting ICs, but the VENTANA PD-L1 (SP142) Assay consistently detects lower proportions of PD-L1-expressing TCs by comparing the same tissue samples (although the staining intensity is very high with this assay). Moreover, while both VENTANA PD-L1 (SP263 and SP142) assays demonstrate a very strong staining intensity, the PD-L1 IHC pharmDx 22C3 Assay often presents with a very weak staining intensity. Variability can be reduced using automated stainers and standardized epitope retrieval steps. Differences in staining characteristics can variably impact the interpretation of PD-L1 positivity. There are also several design specifications that should be considered for the different PD-L1 assays, such as the exclusion of plasma cells from scoring for the VENTANA PD-L1 (SP263) Assay, VENTANA PD-L1 (SP142) Assay, and the PD-L1 IHC pharmDx 22C3 Assay. The approved PD-L1 assays, including technical platforms, scoring algorithms, typical staining characteristics, and special considerations are summarized in Table 1.

Harmonization of PD-L1 testing in UC remains incomplete at present. Currently, the FDA requires, and the EMA recommends, the use of locked assays that have been rigorously validated for use with the manufacturer’s platforms [47,48]. However, interchangeable PD-L1 assays and instruments would provide pathology laboratories with a more practical and cost-effective solution. The results of several studies indicate that the PD-L1 IHC pharmDx 22C3, PD-L1 IHC pharmDx 28-8, and VENTANA PD-L1 (SP263) assays are similar [62]), and concordance studies have shown a high inter-assay correlation between the PD-L1 IHC pharmDx 22C3, PD-L1 IHC pharmDx 28-8, and the VENTANA PD-L1 (SP263) assays [55], with the highest analytical similarity between the VENTANA PD-L1 (SP263) Assay and pharmDx 22C3 Assay [48]. Further validation of interchangeability in larger prospective studies, ideally including samples from patients who have been treated with IO agents, is still needed. Recent data suggest that inter-assay discordances are more likely caused by biological or analytical variables than by antibody epitope [63]. Many pathology laboratories, especially in Europe, are interested in implementing their own laboratory-developed tests (LDTs). In the United States, prescription drug labels stipulate the use of an FDA-approved assay. In Europe, LDTs validated in accordance with the EMA regulations and concordance studies will be required to validate LDTs across institutions utilizing a reference standard. Changing regulation of devices in Europe will increase the requirement for both manufacturers and independent laboratories developing LDTs.

In addition to the proposed practice recommendations (Table 2), appropriate, timely, and ongoing training of pathologists remains very important. If possible, the pathology laboratory should participate in a quality assurance scheme such as the UK National External Quality Assessment Service (UKNEQAS) [64] to ensure best practice. Frequent proficiency testing of pathologists can be used to recognize the “drift” of scoring over time and to reduce inter-reader variation. Pathologists should ideally be trained by the manufacturer or third party and take advantage of available online training tools (i.e., NEQAS, 2019, College of American Pathologists external quality assessment, CAP EQA, 2020 [64,65]). Pathologists should assess the PD-L1 expression following the manufacturer’s “Interpretation Guide” to define a PD-L1 status at the algorithm cutpoint of interest. Regular scoring (e.g., 10 slides per month) maintains practice for a given algorithm and if the practice is limited; refresher training can be taken online. Additionally, the International Quality Network for Pathology (IQN Path) run an academic proficiency testing program for the assessment of pathologists’ and technologists’ accuracy of biomarker readout [66].

## 6. Reporting of Results

We encourage communication between clinicians and pathologists to ensure a fast turnaround and high-quality test result is delivered. Evaluation of the tumor immune response has become a standard part of pathology practice, and diagnostic surgical pathology plays a fundamental role in informing prognosis, guiding IO therapy, and conducting clinical trials (i.e., patient stratification). Therefore, we recommend the PD-L1 test form includes both clinician and pathologist input (Figure 2).

The form allows the pathologist to provide a comprehensive report to the clinician, with PD-L1 status at specific cutpoints, and also the PD-L1 expression raw scores. This informs on eligibility for a pre-specified IO therapy and potential eligibility for alternatives based on the PD-L1 expression. The PD-L1 result should be reported at the cutpoints associated with the therapeutic and a statement regarding eligibility, for example, “SP142, IC > 5%, eligible for atezolizumab.” The TC and IC expression should also be reported separately to allow the clinician to identify other suitable therapies and understand the extent of PD-L1 expression. It is important to remember that the denominators for the scoring algorithms vary and include those shown in Figure 2.

## 7. Emerging Biomarkers in UC

Evidence indicates that PD-L1 by itself may not be sufficient in predicting response to IO agents in metastatic UC [55]. There is growing interest in the potential predictive value of tumor mutational burden (TMB) status in conjunction with PD-L1 expression [43,67,68]. TMB (≥10 mutations per megabase (mut/Mb)) and PD-L1 expression may improve the prediction of response to IO agents in some tumors, such as non-small cell lung cancer [69].

In the PURE-01 study of MIBC, pathologic response to neoadjuvant pembrolizumab was associated with PD-L1–positive (CPS ≥ 10%) or high TMB (median 11.4 mut/Mb) tumors. Notably, there was no correlation between PD-L1 expression and TMB. The data suggest a meaningful cutoff for TMB of ≥15 mut/Mb. Post-therapy reductions in TMB suggest that it may be a biomarker of resistance to IO therapy [29].

In the IMvigor211 trial of metastatic UC following platinum-based chemotherapy, exploratory analyses showed that TMB was an independent predictor of response to atezolizumab [70]. In the first-line atezolizumab study, IMvigor130, TMB (>10 mut/Mb) was predictive for monotherapy benefit (HR 0.71); less so for the combination with chemotherapy (HR 0.82). Survival in PD-L1–high, TMB–high patients was longer (HR 0.22), indicating that a combination of biomarkers can be superior for predicting patient benefit [43]. Comprehensive genomic profiling (whole-exome or whole-genome sequencing) is the gold standard method for TMB measurements, but high costs, long turnaround times, and high tissue sample requirements are limiting factors for routine use in diagnostic laboratories [71].

Targeted next-generation sequencing (NGS) of formalin-fixed, paraffin-embedded blocks is a more feasible approach, and these panels can also provide useful information regarding driver mutations. FDA-approved or authorized diagnostic assays for TMB measurement are the MSK-IMPACT and FoundationOne CDx. The mutational number defining TMB “high” appears to vary across cancer types, and there is unlikely to be a universal number that defines the likelihood of benefit from IO agents in all tumor types. TMB threshold of ≥10 mut/Mb, measured by FoundationOne CDx (equivalent to approximately 200 mutations by whole-exome sequencing) in first-line patients with non-small cell lung cancer was the first established cutoff. Recently, the FDA approved the use of pembrolizumab in patients with TMB ≥ 10 mut/Mb in a pan-tumor approval, which did not include UC. The TMB threshold is 7.4 mut/Mb for the MSK-IMPACT panel and the median number of somatic mutations differs across tumor types. In an analysis of 11,348 patients, microsatellite instability (determined using data from a commercially available NGS panel) was found to be high in 23 of 26 cancer types but not bladder cancer [72]. The overall rates of high microsatellite instability, high TMB, and PD-L1 positivity across all cancer types studied were 3.0%, 7.7%, and 25.4%; only 0.6% of the cases were positive for all three biomarkers [72]. These data, and the results of other studies, suggest the potential need to evaluate more than one individual biomarker to assess the efficacy of IO agents.

Gene expression analyses have been utilized to understand responses to IO therapy better, elucidate mechanisms of resistance, and provide the subclassification of UC tumors. Analyses of gene expression profiles in the T cell-inflamed microenvironment have shown pan-tumor immune-related signatures that correlate with anti–PD-1 benefit, including IFNγ-responsive genes related to antigen presentation, chemokines, cytolytic activity, and adaptive immune resistance [73]. In MIBC, an increase in immune gene expression has been reported in lesions following neoadjuvant treatment with pembrolizumab, including those related to adaptive immunity and innate resistance to anti–PD-1 therapy [29].

UC has a high frequency of somatic mutations, with epigenetic changes and mutations in chromatin remodeling genes being especially frequent [2]. Gene expression profiling has been used to classify UC into different subtypes. Molecular screening using a 64-gene panel has identified three distinct subtypes of MIBC (luminal, basal, infiltrated), predictive of disease-specific survival outcomes [74]. Furthermore, an international consensus statement regarding MIBC subtypes identified six different molecular classes (three luminal subtypes, one stroma rich, one basal, and one neuronal/neuroendocrine-like) [75]. The luminal subtype has been shown to have a poor response to IO agents [70], and a lower immune signature [76], compared with other subtypes. As mutations in the gene encoding the fibroblast growth factor receptor (FGFR) are enriched in the luminal subtype of UC [77], erdafitinib, a tyrosine kinase inhibitor of FGFR1-4, has been approved for the second-line treatment of metastatic UC based on antitumor activity shown in an open-label, phase 2 study [78].

Higher numbers of tumor-infiltrating lymphocytes may be correlated with PD-L1 [79] and predictive of response to IO agents. Moreover, they may be considered to be prognostic in several tumor types and may soon become part of clinical practice/diagnostic pathology; recent evidence suggests that new T-cell clones may enter the tumor to replace pre-existing, exhausted CD8+ T cells in response to IO therapy [80]. Identification of these new T-cell clones in treated tumors may aid in the understanding of responses to IO therapy.

In an analysis of tumors from patients with metastatic UC who had received atezolizumab [81], response to treatment was associated with a CD8+ T-cell phenotype and, to a greater extent, high TMB and lack of response were associated with TGFβ signaling in fibroblasts, which was found predominantly in tumors that showed exclusion of CD8+ T cells from tumor parenchyma. The results suggest that TGFβ limits antitumor immunity with IO agents by restricting T-cell infiltration into tumors. Furthermore, DNA damage response and repair alterations have been demonstrated to be independently associated with response to PD-1/PD-L1 blockade in patients with metastatic UC [82].

Although additional biomarkers, such as TMB, have shown promising results, they are not yet approved for predicting IO response. In the future, other biomarkers may need to be evaluated in conjunction with PD-L1 to improve its predictive utility. A master protocol to include novel biomarkers into the current recommendations for PD-L1 testing may include an integrated report on the available biomarkers of response to IO together with PD-L1 assessment results. This may provide the oncologist/urologist with more insight into how a particular patient will respond to IO. A proposed workflow for evaluation biomarkers in UC may be found in Figure 4.

## 8. Conclusions/Future Directions

Only two IO agents are currently approved for the first-line treatment of cisplatin-ineligible metastatic UC, but there are considerable differences between the companion diagnostic assays for these agents. Biomarker testing is not currently required for the second-line treatment of metastatic UC, despite the approvals of five IO agents. Harmonization of PD-L1 testing remains a need and is increasingly important as the use of IO agents expands to include earlier stages of bladder cancer. Pre-analytical and analytical standards are of great relevance to extract the highest value from PD-L1 assays. The interpretation and reporting of the results of PD-L1 assays require fluid communication between oncologists/urologists and pathologists. PD-L1 expression may change during treatment and/or progression of the disease and may be a mechanism of drug resistance, but PD-L1 testing during the course of treatment is not routinely conducted.

To improve the predictive utility of PD-L1, other biomarkers may need to be evaluated in conjunction with PD-L1, necessitating the development of an integrated master protocol to guide clinicians and pathologists. It is likely that composite biomarker panels may become standard in the future. The challenges involved in developing this new standard will include costs, time, sample availability, and logistics. An additional potential challenge is that the percentage of patients positive using the composite scores may be low, making it harder to identify the most appropriate IO therapy. Further information will be required and will become available as the relationship between PD-L1 and molecular biomarkers is currently being explored in several clinical trials.

The evaluation of combination regimens with IO and non-IO agents will increase the complexity of biomarker testing. Digital pathology, including artificial intelligence (e.g., Flagship Biosciences AI-powered cTA platform), may be utilized to predict response to IO therapy in the future. In this era of personalized medicine, further understanding of PD-L1 and novel biomarkers in UC will help guide patient treatment and IO drug development in the future.

## Figures and Tables

**Figure 1 cancers-13-01424-f001:**
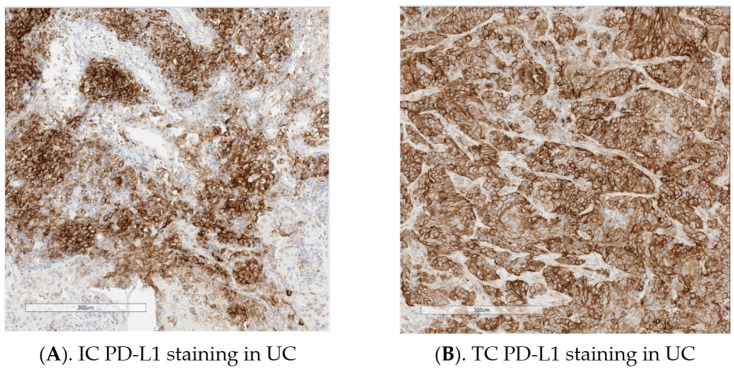
(**A**,**B**). Examples of immune cell (IC) and tumor cell (TC) programmed cell death-ligand 1 (PD-L1) staining in urothelial carcinoma (UC) using the VENTANA PD-L1 (SP263) Assay (Images provided by Marietta Scott, Precision Medicine & Biosamples, AstraZeneca).

**Figure 2 cancers-13-01424-f002:**
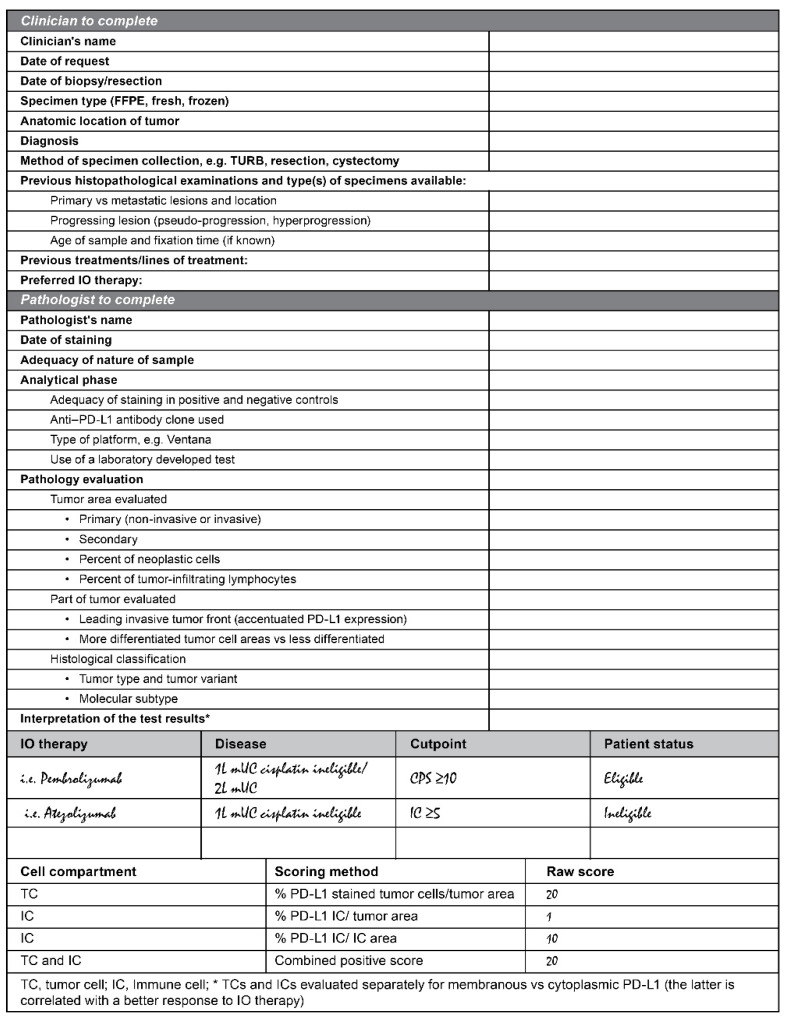
Example of a PD-L1 test request form.

**Figure 3 cancers-13-01424-f003:**
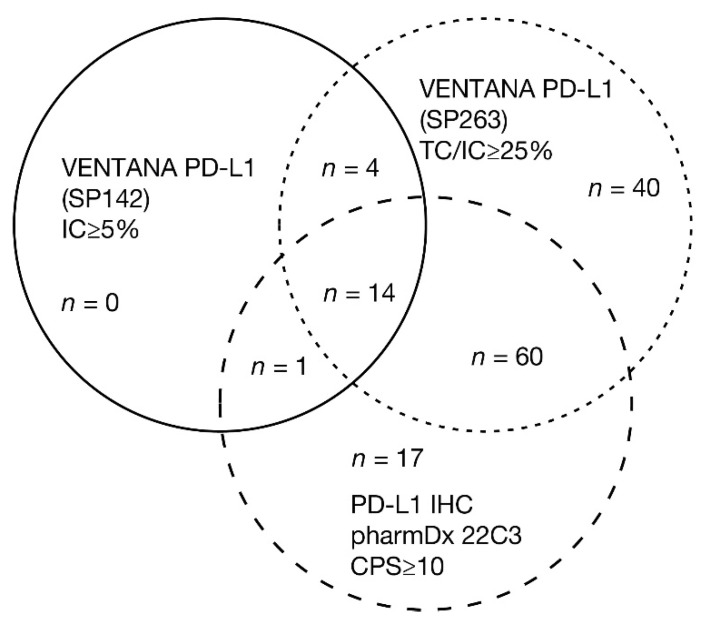
UC specimens classified as PD-L1 high using commercially available PD-L1 assays. The figure shows the overlap of positive cases using three commercially available PD-L1 assays. In an assay comparison study, 335 UC tumor samples were stained and scored with the algorithm and cutpoint associated with that assay [48]. Of the 40% of samples deemed PD-L1 high using any assay, 12% were uniquely identified by the VENTANA PD-L1 (SP263) Assay, 5% by the PD-L1 IHC pharmDx 22C3 Assay, and none were unique to the VENTANA PD-L1 (SP142) Assay. The highest overlap in assays was observed between the VENTANA PD-L1 (SP263) Assay and PD-L1 IHC pharmDx 22C3 Assay, where 22% of samples were PD-L1 high with either assay. If extrapolated to a clinical setting, this exploratory study indicates that of 100 patients tested with all three assays, only four patients would be deemed PD-L1 high with all three assays (with acknowledgment to Marietta Scott, Precision Medicine, AstraZeneca, Cambridge, UK, for analysis of the data).

**Figure 4 cancers-13-01424-f004:**
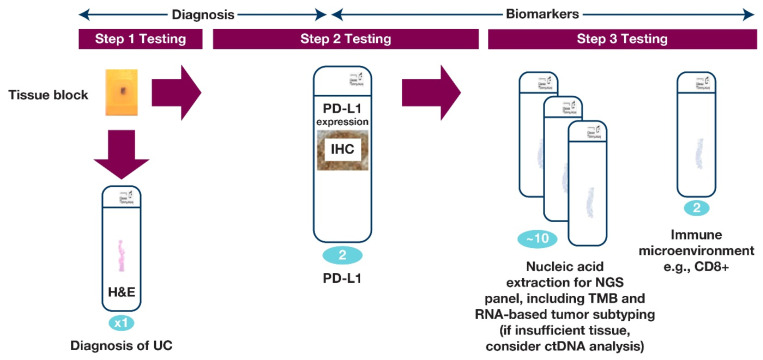
UC biomarkers workflow, which is an integrated platform to report PD-L1 and other biomarkers. Figure 4 illustrates a prospective integrated protocol for multiple biomarker testings on samples from patients with UC. The numbers of sections required are shown in blue. Ideally, in the future, step 1 testing would be combined with step 2 testing to accelerate information to the clinician and inform potential treatment. A minimum of two slides are required for PD-L1 testing and tissue requirements for next-generation sequencing (NGS) may vary according to the testing platform used.

**Table 1 cancers-13-01424-t001:** Scoring algorithms for the companion diagnostic assays of immuno-oncology (IO) agents.

Anti–PD-1/PD-L1 Therapeutics	Atezolizumab	Avelumab	Durvalumab, Avelumab, and Tislelizumab	Nivolumab	Pembrolizumab
Assay	VENTANA PD-L1 (SP142) Assay [52]	PD-L1 IHC pharmDx 73-10 [not FDA approved]	VENTANA PD-L1 (SP263) Assay [53]	PD-L1 IHC pharmDx 28-8 [50]	PD-L1 IHC pharmDx 22C3 [51]
Scoring algorithm	IC ≥ 5%; number of PD-L1–positive tumor-infiltrating ICs as a proportion of the total TC and IC area	TC ≥ 5%; number of PD-L1–positive TCs as a proportion of the total TC area	TC or IC ≥ 25%; number of PD-L1–positive TCs with membrane staining as a proportion of the total TC area or PD-L1–positive ICs with membrane, cytoplasm, or punctate as a proportion of the total IC area.	TC ≥ 1%; number of PD-L1–positive TCs as a proportion of the total TC area	Number (Count) of PD-L1–positive TCs and number of PD-L1–positive ICs as a proportion of the total TC area
Typical staining characteristics	Dot-/ant-like staining pattern Low tumor cell staining Strong IC staining Developed for immune cell scoring		Homogenous tumor cell staining Mostly strong staining intensity for TCs and ICs	Homogenous tumor cell staining Moderate-strong staining intensity	Homogenous tumor cell staining Mostly weak staining intensity
Design Considerations	Plasma cells have to be excluded from scoring All immune cells are included (incl. neutrophil granulocytes)		Immune cell positivity is scored according to the area occupied by all immune cells (IC-“Area”-score) TC and IC are scored independently. Patients are positive when exceeding one of the two cutoffs or both PD-L1 can also be considered high if: ICP * > 1% and IC+ ≥ 25%; or, ICP * = 1% and IC+ = 100%. Plasma cells have to be excluded from scoring All immune cells are included (incl. neutrophil granulocytes)		Combined positive score including immune cells and tumor cells Plasma cells have to be excluded from scoring Neutrophil granulocytes not included

* ICP, tumor-associated immune cells in the tumor area.

**Table 2 cancers-13-01424-t002:** Recommendations for optimal conditions for PD-L1 immunohistochemistry.

Specimen Selection	Recommendations for Optimum Conditions *
Site of biopsy	Use biopsies from a sample relevant to the disease stage, e.g., TURBS for MIBC, primary/metastatic for mUC.
Specimen age	Use formalin-fixed, paraffin-embedded sample blocks which have not been subjected to warm or fluctuating temperatures. Use the most recent sample proximal to starting therapy (maximum of 1 year old) [55]. When using approved assays follow the manufacturer’s instructions as cut slide stability can range from 1 to 6 months (PD-L1 IHC pharmDx 22C3 PI [51] and VENTANA PD-L1 (SP263) PI [58]).
Specimen type	Score the whole slide. Select a tissue specimen with invasive disease. ** TURB samples may be considered, however, only if they contain the invasive disease. Avoid highly necrotic samples where possible. Use positive controls for PD-L1; lymphatic tonsil tissue is recommended as optimal, with positive staining for macrophages, dendritic cells, and lymphocytes. Do not use cytology/smears for scoring ICs, as tissue architecture is necessary to understand if ICs are tumor infiltrating. It is currently not generally advised to use samples of bone metastases. There is however some evidence that bone decalcification using EDTA solution may yield good results and could possibly be used for PD-L1 IHC. Validation is required.
Sample preparation/fixation	Use 10% neutral buffered formalin in a quantity > 10 times the volume of the specimen. Sample should be placed in formalin as soon as possible (<30 min) and for a period of 12–24 h. Longer fixation may cause diffuse staining patterns. For immunohistochemistry, fixation should be performed for a minimum of 6 h and no more than 72 h. Large samples should be excised to allow for sufficient penetration of the fixative. Fixative penetrates about 1 mm/h with slight variation across different types of tissues. i.e., for cystectomy samples, the tumor should be excised, or the bladder opened to allow fixative to penetrate. Avoid decalcified tissue or tissue processed with other fixatives. Use on-slide positive and negative controls from the same institution or manufacturer, especially if an automated stainer is not used. Section specimens into a thickness of 3 or 4 µmm (as specified in manufacturers instruction).

* Author recommendations only; please refer to the manufacturers’ package insert instructions for the licensed use (VENTANA PD-L1 (SP263) Assay PI [58] and VENTANA PD-L1 (SP142) Assay PI [57]). ** As most IO agents are currently only approved for locally advanced or metastatic UC (except for pembrolizumab, which is also approved for non–muscle-invasive bladder cancer in the USA), we recommend selecting a tissue specimen with invasive disease, rather than TURB.

## Data Availability

Data underlying the findings described in this manuscript may be obtained in accordance with AstraZeneca’s data sharing policy described at https://astrazenecagrouptrials.pharmacm.com/ST/Submission/Disclosure accessed on 19 March 2021.

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
