# Peer review of "Immune Checkpoint Inhibitors in Urothelial Carcinoma: Recommendations for Practical Approaches to PD-L1 and Other Potential Predictive Biomarker Testing"

_cancers, 2021, doi:10.3390/cancers13061424_

Round 1

Reviewer 1 Report

The authors described current status and problems in pathological evaluation of PD-L1 status and immune microenvironment of UC tissues. The authors also provided recommendations for optimal specimen processing for PD-L1 testing and proposed an example of PD-L1 test request form which contains required information for both pathologists and clinicians.    This is a well-written, comprehensive narrative review and experts’ recommendation based on international workshops held by expert pathologists. This paper would be useful for not only pathologists and clinicians but also technicians and pharmacists involved in PD-L1 testing and treatment with anti-PD1 or PD-L1 antibodies.   The authors may want to address following points before publication. 1_ Previous publications indicate PD-L1 testing is useful for predicting response to anti-PD1/PD-L1 antibodies in NSCLC but not in UC (and RCC). The reviewer wonders what are different between NSCLC and UC. The authors may want to make some comments on this important issue.   The other points include: 2_ Please spell-out “CE” at its first appearance (line 97). 3_ Lines 351-353: Please clarify whether the description is about UC or all types of tumor.

Reviewer 2 Report

This is an interesting and high profile paper concerning the (potential) role of immune checkpoint inhibitors in bladder cancer.
It is well written and easily readable. Anyway some considerations emerges.
It is difficult to consider the paper a strict review of the topic (at least a mini-review). The Authors considered some recent workshops and mainly around  these they made their considerations. On the other hand the manuscript is not a classically structured work. So, taking in mind these points, we have to consider the manuscript a narrative work which pointed out some aspects of immune checpoint inihibitors in view of a more narrow collaboration among pathologists and clinicians. If we read in this way the paper, I think it is a good manuscript.

Reviewer 3 Report

The work presented by the authors is important for clinical practice decisions in IO for MIBC.

There were three workshops organised. Were the authors the only participants or were other people involved?

During ASCO GU several studies with contradicting data were presented concerning PDL-1 expresion and treatment outcome. Do the authors have an explanation?

What about PD1 and PDL1 inhibitors and level of PDL1 expression and outcome?

Should we adapt our treatment to levels of expression?

Author Response

We want to point out that since comments from Reviewer #3 were entered into the system on March 5th, after the authors received the journal notification on March 4th, we have not provided a response at this time in order to meet the extended deadline of Friday, March 12th. However, we would be happy to address these additional new comments by Friday, March 19th if required.

Reviewer 4 Report

This manuscript gives a good overview on PDL1 testing platforms in UCB. There is ongoing controversy on PDL1 status testing in UCB patients and it is still a rapid evolving field of uro-oncology. Therefore, the present manuscript is of interest.

Some comments:

Please include a statement on the possible impact of AstraZeneca on the intelectual content of the manuscript.

„....remains today as the standard of care (SoC) for non muscle-invasive disease [3].“

Please add: in patients with highrade tumors and/or T1 tumors.

Please include also that Mitomycin is a treatment option in Ta highgrade tumors.

Please comment on inter/intraobserver variability of results of the different available PDL1 platforms.

It would be great for the readers to include some clinical cases (PDL1 low expression vs. PDL1 high expression; with images etc.). Would that be possible?

Please comment on the possibilty of evaluating PD1/PDL1 of circulating tumor cells in UCB.
